# Emerging Therapeutic Approaches and Genetic Insights in Stargardt Disease: A Comprehensive Review

**DOI:** 10.3390/ijms25168859

**Published:** 2024-08-14

**Authors:** Laura Andreea Ghenciu, Ovidiu Alin Hațegan, Emil Robert Stoicescu, Roxana Iacob, Alina Maria Șișu

**Affiliations:** 1Department of Functional Sciences, “Victor Babeș” University of Medicine and Pharmacy Timișoara, Eftimie Murgu Square No. 2, 300041 Timișoara, Romania; bolintineanu.laura@umft.ro; 2Discipline of Anatomy and Embriology, Medicine Faculty, Vasile Goldis Western University of Arad, Revolution Boulevard 94, 310025 Arad, Romania; 3Field of Applied Engineering Sciences, Specialization Statistical Methods and Techniques in Health and Clinical Research, Faculty of Mechanics, ‘Politehnica’ University Timișoara, Mihai Viteazul Boulevard No. 1, 300222 Timișoara, Romania; stoicescu.emil@umft.ro (E.R.S.); roxana.iacob@umft.ro (R.I.); 4Department of Radiology and Medical Imaging, ‘Victor Babeș’ University of Medicine and Pharmacy Timișoara, Eftimie Murgu Square No. 2, 300041 Timișoara, Romania; 5Research Center for Pharmaco-Toxicological Evaluations, ‘Victor Babeș’ University of Medicine and Pharmacy Timișoara, Eftimie Murgu Square No. 2, 300041 Timișoara, Romania; 6Department of Anatomy and Embriology, ‘Victor Babeș’ University of Medicine and Pharmacy Timișoara, 300041 Timișoara, Romania; sisu.alina@umft.ro

**Keywords:** Stargardt disease, *ABCA4* gene, gene therapy, stem cell therapy, pharmacological interventions, CRISPR/Cas9, precision medicine

## Abstract

Stargardt disease, one of the most common forms of inherited retinal diseases, affects individuals worldwide. The primary cause is mutations in the *ABCA4* gene, leading to the accumulation of toxic byproducts in the retinal pigment epithelium (RPE) and subsequent photoreceptor cell degeneration. Over the past few years, research on Stargardt disease has advanced significantly, focusing on clinical and molecular genetics. Recent studies have explored various innovative therapeutic approaches, including gene therapy, stem cell therapy, and pharmacological interventions. Gene therapy has shown promise, particularly with adeno-associated viral (AAV) vectors capable of delivering the *ABCA4* gene to retinal cells. However, challenges remain due to the gene’s large size. Stem cell therapy aims to replace degenerated RPE and photoreceptor cells, with several clinical trials demonstrating safety and preliminary efficacy. Pharmacological approaches focus on reducing toxic byproduct accumulation and modulating the visual cycle. Precision medicine, targeting specific genetic mutations and pathways, is becoming increasingly important. Novel techniques such as clustered regularly interspaced palindromic repeats (CRISPR)/Cas9 offer potential for directly correcting genetic defects. This review aims to synthesize recent advancements in understanding and treating Stargardt disease. By highlighting breakthroughs in genetic therapies, stem cell treatments, and novel pharmacological strategies, it provides a comprehensive overview of emerging therapeutic options.

## 1. Introduction

Stargardt disease, also known as Stargardt macular dystrophy or juvenile macular degeneration, was initially described in 1909 by German ophthalmologist Karl Stargardt, who reported the disease in seven patients from two families who had macular degeneration with yellow–white pisciform flecks surrounding it [1]. It is the most common form of inherited juvenile macular degeneration, with an estimated prevalence of 1 in 8000 to 10,000 individuals. The disease affects both males and females equally and can occur in various ethnic groups. The onset of symptoms typically occurs in childhood or adolescence, although the rate of progression can vary widely among individuals [2].

The primary risk factor for Stargardt disease is genetic. The most commonly implicated gene is ATP-binding cassette, sub-family A, member 4 (*ABCA4*), which plays a crucial role in the visual cycle by transporting toxic substances away from photoreceptor cells in the retina. The *ABCA4* gene yields a transmembrane protein, which is part of the ATP-binding cassette transporter superfamily and is crucial for retinoid recycling in the visual cycle and is found on disk membranes in the outer segments of cones and rods. The absence of *ABCA4* protein results in the buildup of visual cycle byproducts (lipofuscin), malfunction of the retinal pigment epithelium (RPE), and consequent photoreceptor cell degradation [3]. Other genes, such as the elongation of very-long-chain fatty acids protein 4 (ELOVL4) and prominin-1 (PROM1), have also been associated with the disease, although they are less common [2].

Currently, there is no cure for Stargardt disease, and management focuses on supportive measures to maximize remaining vision and quality of life. Low vision aids, such as magnifiers, help patients utilize their peripheral vision [4]. Over the last few years, there has been an increase in studies on the clinical and molecular genetics of Stargardt disease. This has aided in a better knowledge of the basic pathophysiology, resulting in both finished and active trials, as well as a diverse set of prospective clinical trials. A variety of strategies have been investigated for managing Stargardt disease, including pharmaceutical therapies, stem cell therapies, and gene therapy. Precision medicine, which focuses on specific variations and pathways, is becoming increasingly popular [1,4,5].

Stargardt disease poses significant public health challenges due to its impact on the quality of life and the burden it places on affected individuals and their families [6,7]. Vision loss can lead to difficulties in reading, recognizing faces, and performing everyday tasks, which can affect educational and occupational opportunities. The early onset of the disease means that affected individuals may require long-term support and rehabilitation services. The psychological impact of progressive vision loss, especially in young people, can also lead to emotional and social challenges [8]. Therefore, awareness and early detection are crucial to provide appropriate support and resources. From a public health perspective, genetic counseling plays a vital role in helping families understand the inheritance patterns and risks of Stargardt disease [2,5,8]. Public health initiatives should focus on educating the public and healthcare providers about the disease, promoting research for potential treatments, and providing resources for affected individuals and their families.

Our focus is on Stargardt disease (STGD1), particularly emphasizing the investigation of novel therapeutic approaches. This includes exploring advanced genetic therapies, stem cell treatments, and innovative pharmacological interventions that aim to halt or reverse the progression of vision loss associated with this condition. By concentrating on the latest developments and cutting-edge research, we aim to provide a comprehensive overview of potential treatments that could offer hope for patients with Stargardt disease.

## 2. Disease Overview

Initially defined as a macular dystrophy, Stargardt disease (OMIM 248200), sometimes referred to as STGD1, constitutes one of the most prevalent genetically acquired retinal diseases, causing approximately 12% of inherited retinal disease-related blindness [9]. Many gene alterations are at the base of inherited retinal diseases (IRDs), impacting rod, cone, and RPE functions. More than 400 mutated genes have been identified as responsible for IRDs, which includes Leber congenital amaurosis, retinitis pigmentosa (RP), Usher syndrome, Stargardt disease, and cone–rod dystrophies, among others [10]. There are several phenotypic and genotypic variants of Stargardt disease, the most prevalent of which is known as STGD1. Rarely, a condition known as Stargardt-like disease may develop; this condition is frequently classified as Stargardt disease. Autosomal dominant mutations in ELOVL4 (STGD3) or PROM1 (STGD4), the genes encoding the elongation of very-long-chain fatty acids and prominin 1, respectively, are the root cause of Stargardt-like disease [11]. *ABCA4* is the source of biallelic mutations that cause an autosomal recessive IRD [12]. Despite the significant allelic variability of *ABCA4*, which has been linked to over 1200 disease-causing variants to date, founder mutations linked to STGD1 have been detected in a number of demographic categories, including Europeans, Afro-Americans, and Asians [13]. Despite being the most prevalent cause of juvenile macular dystrophy, Stargardt disease is predicted to affect 10 to 12.5 people per 100,000 in the United States of America [1]. According to Cornish et al., the yearly incidence of this disorder in the United Kingdom is between 0.11 and 0.12 per 100,000 people [14]. According to a study conducted in the Netherlands, there are 1.67 to 1.95 cases of Stargardt illness in every 1,000,000 people annually [15]. STGD1 patients usually begin presenting with evolving painless bilateral central vision impairment, with visual acuity between 20/20 and 20/400, as well as symptoms such as central scotoma, prolonged dark adaptation, light sensitivity, photopsia, and inappropriate color vision, which affects both males and females. It is now widely acknowledged that the disease’s spectrum may range from childhood-onset cone–rod dystrophy with quickly developing central and peripheral visual impairment to late-onset macular pattern dystrophy-like disease that has a tendency to preserve the fovea [16]; nevertheless, the age of onset can also vary within the same family [17]. Advanced imaging techniques, including fundus photography (Figure 1 and Figure 2) and autofluorescence (FAF), optical coherence tomography (OCT), and fluorescein angiography (FA), provide detailed views of retinal structure and function, highlighting areas of retinal atrophy and lipofuscin accumulation. Disease advancement rates can be measured by determining the rate at which the area of RPE atrophy expands in near-infrared and short-wavelength FAF images [18], measuring the strength of autofluorescence signal in regions unaltered by flecks or RPE loss [19], and counting hyperautofluorescent flecks [20]. Genetic testing is crucial for confirming the diagnosis by identifying mutations in the *ABCA4* gene. Whole-genome sequencing identifies up to 95% of causative mutations in clinically well-characterized STGD1 patients [12]. Quality of life is often compromised due to the associated visual limitations and the emotional strain of coping with a chronic condition. However, Stargardt disease does not generally influence life expectancy, allowing patients to live a normal lifespan despite vision-related challenges.

## 3. Function of Retinal Pigment Epithelium and Photoreceptor Cells

The retinal pigment epithelium (RPE) is a monolayer of pigmented cells located between the neurosensory retina and the choroid, crucial for retinal health and visual function. RPE cells are essential in phagocytosing photoreceptor outer segment (POS) disks, aiding in photoreceptor renewal, and they play a key role in the visual cycle by recycling all-trans-retinal to 11-cis-retinal for phototransduction [21,22,23]. These cells also regulate the exchange of nutrients, ions, and waste between the retina and choroid, absorb scattered light to reduce phototoxicity [22], and secrete growth factors like vascular endothelial growth factor (VEGF) and pigment epithelium-derived factor (PEDF), which support the choriocapillaris and photoreceptors. Additionally, RPE cells contain melanin and antioxidants that protect the retina from oxidative stress [21].

The RPE serves a function not solely for sustaining retinal homeostasis but also for neuro-induction. It produces a variety of growth factors that help retinal cells differentiate and regenerate, including photoreceptors. The findings show that RPE-conditioned media can trigger neural development in adipose-derived mesenchymal stem cells, resulting in the release of neuronal and glial markers. This shows that RPE cells can drive neural development beyond photoreceptors, extending to adult stem cells, showing their larger neuro-inductive capacity [24]. Furthermore, studies have demonstrated that RPE cells impact the growth and differentiation of photoreceptors and other retinal cell categories via neurotrophic factor production. These substances have an important role in the survival and development of retinal progenitors and adult stem cells, emphasizing the RPE’s crucial neuro-inductive properties in retinal therapeutics [25].

Photoreceptor cells are specialized neurons that convert light into electrochemical signals. Rods are highly sensitive to photons and capable of detecting single photon events, but they do not mediate color vision and have low spatial acuity. Rods contain the photopigment rhodopsin, which undergoes a conformational change when exposed to light, initiating a phototransduction cascade that hyperpolarizes the rod cell and generates a neural signal [23] (Figure 3). Cones mediate high-resolution central vision and color discrimination. The human retina contains three types of cones, each with a different photopigment sensitive to different wavelengths of light: S-cones (short wavelengths, blue), M-cones (medium wavelengths, green), and L-cones (long wavelengths, red). These photopigments, called opsins, similarly undergo a conformational change upon light absorption, triggering a phototransduction cascade specific to cones [26].

In the phototransduction process, light absorption by photopigments leads to the activation of transducin, a G-protein, which subsequently activates phosphodiesterase (PDE). PDE hydrolyzes guanosine 3′, 5′-cyclic monophosphate (cGMP), leading to the closure of cGMP-gated ion channels in the photoreceptor outer segment membrane. This results in the hyperpolarization of the photoreceptor cell and a reduction in neurotransmitter release at the synapse with bipolar cells, ultimately modulating the visual signal transmitted to the brain via the optic nerve [27].

In Stargardt disease, mutations in the *ABCA4* gene disrupt the normal function of RPE cells through several mechanisms. These mutations lead to defective transport of all-trans-retinal, resulting in its accumulation within photoreceptor cells. This buildup causes the formation of toxic bisretinoids, including A2E, which are deposited in the RPE cells as lipofuscin. The excessive accumulation of lipofuscin impairs the cellular functions of RPE cells [28]. The presence of A2E and other bisretinoids increases the susceptibility of RPE cells to oxidative stress, as lipofuscin granules generate reactive oxygen species (ROS) upon light exposure, leading to oxidative damage and the apoptosis of RPE cells.

This accumulation also interferes with the RPE cells’ ability to phagocytose and degrade photoreceptor outer segments efficiently, disrupting the essential renewal processes and leading to secondary degeneration of photoreceptors. Chronic accumulation of toxic byproducts, oxidative stress, and impaired cellular functions ultimately result in the death of RPE cells. As RPE cells degenerate, they fail to support the overlying photoreceptors, leading to their progressive degeneration and the characteristic central vision loss seen in Stargardt disease [29].

The macula, with its high density of cone photoreceptors responsible for sharp, central vision, is particularly affected in Stargardt disease. As RPE cells degenerate and fail to support the overlying photoreceptors, cone cells in the macula begin to deteriorate, resulting in progressive central vision loss [30]. This degeneration of cones manifests clinically as difficulty with tasks requiring detailed vision, such as reading and recognizing faces.

The long-term objective of studies on human retinal degenerative changes is eyesight rehabilitation. On the other hand, the quantity of preserved photoreceptors determines the success rate. For human gene therapy to be successful, retinal regions containing retained photoreceptors must be identified and targeted. Therefore, in order to find appropriate candidates for gene-based therapeutic applications, comprehensive noninvasive clinical examinations on individuals with *ABCA4* mutations are necessary [29].

## 4. Novel Therapies in Stargardt Disease

There is no agreement on the best treatment option for patients with Stargardt disease, and therapy options are limited. Recent research has concentrated on improving understanding of the underlying genetic abnormalities and pathogenetic pathways that cause visual loss in STGD1, rather than developing guidelines for follow-up and treatment. Even though there is no approved treatment for this disease, several therapies are being investigated.

### 4.1. Gene Therapy

With numerous experimental studies currently underway and an authorized gene therapy treatment for hereditary retinal degeneration (Table 1), evidence of the method’s safety and efficacy is growing year after year [30,31]. Presently, the emphasis has been on gene supplementation procedures for genes that trigger autosomal recessive and X-linked disorders which are naturally receptive to such a strategy. STGD1 is a promising target for this type of treatment; nevertheless, the *ABCA4* gene’s long 6.8 kb coding sequence presents obstacles to gene delivery. Identifying an effective vector for delivering the *ABCA4* gene to the retina has proven challenging due to its large size and location. To deliver *ABCA4* protein to the photoreceptors, vectors must be capable of transducing this cell layer. The recommended vector, adeno-associated virus (AAV), has an ideal packaging capacity of roughly 4.7 kb. This constraint has prompted the creation of a variety of ways for delivering large genes [11]. On the other hand, the coding sequences for ELOVL4 and PROM1 are appropriately sized for AAV manufacturing. However, because these genes are connected with autosomal dominant disorders and dominant negative processes [32], gene supplementation by itself will probably fail to treat the disease. As a result, due to the absence of successful pharmaceutical therapies, alternative treatments for STGD3 and STGD4 will most likely rely on further developments in gene-editing technology [11]. As with any other category of IRD, the selected treatment method for *ABCA4*-associated retinopathy may vary from mutation-specific techniques to more commonly used cell replacement, primarily dependent on the initial genetic abnormality and the stage of the disease at the beginning of therapy (Figure 4).

#### 4.1.1. Lentiviral Vectors

Lentiviral vectors, with a packaged capacity of ~8 kb, can carry even the biggest coding sequences, such as *ABCA4*, making them promising for gene therapy. Unlike other retroviruses, lentiviruses may penetrate non-dividing cells by importing their pre-integration complex into the nuclear compartment [10]. A group of scientists evaluated the efficacy and biodistribution of a vesicular stomatitis virus (VSV)-G pseudotyped equine infectious anemia virus (EIAV)-based lentiviral vector harboring a coding sequence for the *ABCA4* gene when injected subretinally. The toxicological impact of lentiviral vectors (LV) was investigated in rabbits and non-human primates, and there were no significant variations in mortality, body and organ weights, tissue modifications, or blood changes between the treatment and control groups. This investigation established the LV’s tolerability and localized expression in the target area. It offered assistance to the start of StarGen’s first-in-man research study [33]. StarGen launched a Phase I/IIa clinical trial in 2011 (SAR422459, NCT01367444), which concluded in 2019. This clinical experiment was designed to evaluate the vector’s efficacy as well as potential benefits in eyesight and structural modifications [34]. These trials found that the EIAV vector is generally tolerated effectively, with patients experiencing one or more (non-serious) adverse events; however, autofluorescence changes in nearly one-third of patients call for more inquiry into the vector’s safety. The actual reasons for this experiment’s termination are unknown; however, another study using the same compounds is still underway and recruiting. On the other hand, Parker et al. [35] published a test–retest variability for several clinical outcomes in participants of this trial.

The absence of efficacy evidence from research studies lends little credibility to the ongoing application of lentiviral vectors in Stargardt disease gene therapy. Over the last years, the popularity of lentiviral vectors has diminished, whereas use of AAV vectors has increased dramatically, indicating a move toward more promising and effective gene delivery strategies for this illness.

#### 4.1.2. Adeno-Associated Viral Vectors

The AAV vector is one of the most actively investigated tools in gene therapy, due to its ability to deliver genetic material safely and efficiently into target cells [36]. AAV vectors are non-pathogenic and have a low immunogenic profile, making them ideal for long-term gene expression. They can infect a variety of cell types, both dividing and non-dividing, and are currently being explored for the treatment of numerous genetic disorders. Their optimal packaging capacity is around 4.7 kb, which poses limitations for delivering larger genes but has driven the development of innovative strategies to overcome this challenge. Preclinical and clinical breakthroughs in AAV-mediated gene replacement and gene editing have contributed to AAV’s reputation as an excellent pharmaceutical vector, with two AAV-based therapies receiving regulatory clearance in Europe or the United States [36]. Several dual AAV approaches may effectively transport large genes to diverse tissues, including trans-splicing, overlapping, and hybrid dual vector approaches. Research has demonstrated that messenger ribonucleic acid (mRNA) transcripts that underwent splicing may have greater translational yields than identical intronless transcripts. The findings of McClements et al. support those of prior research, indicating that introns are commonly used in vector transgenes [37]. *ABCA4* expression was successfully achieved in adult *ABCA4*−/− mice using dual vector approaches [38] and nanoparticle delivery [39], indicating beneficial benefits. The study of McClements is the first to demonstrate *ABCA4* expression in the photoreceptor outer segments of mature *ABCA4*−/− mice retinae after injections with an overlapping dual vector system. Two companies (Splice Bio (Princeton, NJ, USA) and AAVantgarde Bio, Milano, Italy) are using dual viral vector technology and intend to launch Phase I/II gene therapy clinical trials in 2024–2025 (https://splice.bio/pipeline/; accessed on 26 June 2024; https://www.aavantgardebio.com/technology-pipeline/; accessed on 15 June 2024). Ocugen reported that the second group of their Phase 1/2 GARDian clinical trial for OCU410ST, a modifier gene therapy candidate under study for Stargardt disease, has completed dosing as a one-time treatment for life. The GARDian clinical trial is a multicenter study that will be undertaken in two phases and will enroll up to 42 individuals, employing the adeno-associated viral vector 5 human RORA (NCT05956626).

#### 4.1.3. Other Genetic Approaches

Nanoparticles provide an alternate carrier for larger transgenes, perhaps bypassing the immunological response induced by viral vectors in inherited retinal disease. The first nanoparticles employed for *ABCA4* transportation, polyethylene glycol-substituted polylysine (CK30PEG), had a packing capacity of 5–20 kb. Mice were given subretinal injections of nanoparticles with plasmids harboring the entire *ABCA4* transgene, which was driven by either the interphotoreceptor retinoid binding protein (IRBP) or mouse opsin promoter. This resulted in fewer lipofuscin granules in the RPE than in untreated eyes. These findings indicate that nanoparticles can efficiently transport plasmid deoxyribonucleic acid (DNA) to photoreceptors for a lasting treatment expression [40]. Sun et al. used self-assembled nanoparticles of (1-aminoethyl)iminobis[N-(oleicylcysteinyl-1-amino-ethyl)propionamide] (ECO) and an *ABCA4* plasmid with a bovine rhodopsin promoter to increase gene expression in retinal cells. ECO/pRHO-ABCA4 therapy significantly reduced A2E accumulation (by 35%) in *ABCA4*−/− rodents 6 months after only one injection, compared to a control group. In comparison, ECO/pCMV-*ABCA4* only reduced A2E by around 15%. Future research is needed to improve the ECO/pDNA nanoparticles to raise *ABCA4* expression and enhance their therapeutic effectiveness. Although it may be difficult for ECO/pRHO-ABCA4 to produce a substantial amount of *ABCA4* expression, as found in the retina of rodents, improved gene expression via this non-viral strategy would result in a lower A2E accumulation [41].

In certain forms of Stargardt disease, there are no errors in the portions of the *ABCA4* gene that encode for the structural blocks of the *ABCA4* protein. A standard genetic test only evaluates a portion of each gene, excluding non-coding portions referred to as introns. These portions are modified during protein development via a process identified as splicing. Nevertheless, genetic errors within introns can still substantially impact how the coding regions are altered and processed by the cells, frequently resulting in a defective protein. A standard genetic test only evaluates a portion of each gene, excluding non-coding portions referred to as introns. These portions are modified during protein development via a process identified as splicing. Nevertheless, genetic errors within introns can still substantially impact how the coding regions are altered and processed by the cells, frequently resulting in a defective protein [11]. 

Various mutation-specific treatments are being developed for *ABCA4*-associated retinopathy. These tactics mostly use antisense oligonucleotides (AONs) and target variations that influence *ABCA4* pre-mRNA splicing. AONs are short and adaptable RNA molecules that can be generated with a sequence that complements the target pre-mRNA. So far, AONs have been mostly employed to prevent PE inclusions induced by deep-intronic alterations [11].

Genome sequencing can detect a significant number of noncoding sequence variants. Still, it can be difficult to determine the causal variants with no assays that highlight the associated mRNA and/or protein abnormalities [42]. Scientists have devised a rapid and cost-effective approach for scanning the complete length of the *ABCA4* gene, which involves the introns. This innovative technique has allowed them to uncover multiple intronic defects, and they proceeded to construct a form of molecular patch, characterized as a “band-aid” to counteract the damaging consequences of these intronic genetic errors, resulting in normal protein being build-up again. Such ‘genetic editing’ technologies are especially useful for intronic genetic defects. However, because each genetic flaw has its unique band-aid, acquiring a genetic diagnosis will be necessary in order to potentially profit from such therapies in the near future. In the study of Sangermano et al. [42], they found deep-intronic variants in 67% of STGD1 individuals with no *ABCA4* variants, one variant, or one causative variant in trans with c.5603A > T. AONs targeting a specific pseudoexon has dramatically improved the splicing deficiency reported in the photoreceptor precursor cells of patients with mutations [43]. Other studies have also shown that employing midigene splice assays and patient-derived cells, the capacity of AONs to avoid the abnormal PE inclusions induced by numerous distinct *ABCA4* variations has been proven, albeit the effectiveness of therapy has only been examined at the RNA level [44,45].

Ascidian Therapeutics is developing ‘exon editing’ as another unique gene modification technique, with a Stargardt treatment study scheduled in the near future. Unlike the previously mentioned ‘band-aid’ strategy, which would necessitate the development of many particular ‘band-aids’, this treatment could potentially be applicable to all genetic defects causing Stargardt Disease (https://ascidian-tx.com; accessed on 28 May 2024).

SGT-1001 aims to treat the primary genetic source of Stargardt disease. It is composed of up of a full-length *ABCA4* gene construct and mRNA that encodes the SaliogaseTM enzyme, SalioGen’s patented mammalian-derived bioengineered enzyme. SGT-1001 uses a patented lipid nanoparticle that is administered subretinally and only needs to be administered once. Preclinical experiments reveal significant gene integration in PR and RPE cells, as well as adequate *ABCA4* expression to lower lipofuscin A2E levels associated with retinal degeneration in a validated animal efficacy model. SalioGen intends to start clinical trials in the first half of 2025.

Optogenetics is a genetic strategy for converting non-light-sensitive cells in the retina into light-sensitive cells. Several various ‘light switches’ and techniques are being investigated, including an ongoing Phase I/II study in Stargardt disease (Nanoscope Therapeutics, Dallas, TX, USA). STARLIGHT is a Phase 2 trial (NCT05417126) evaluating the safety and effects of a single intravitreal injection of virally-carried Multi-Characteristic Opsin (vMCO-010) in patients with Stargardt Disease. Raytherapeutics is also in the initial phase of developing a treatment (RTx-021), with a Phase I clinical trial being expected to start soon (https://raytherapeutics.com/pipeline/; accessed on 2 June 2024). 

Vectors with high packaging capacity, like lentiviruses, adenoviruses, and nonviral vectors, may not adequately transduce adult photoreceptors. However, vectors based on AAV viruses, which have been employed in clinical trials for ocular illnesses, have a typical packing capacity of roughly 4.7 kb, making them insufficient to package the massive *ABCA4* coding sequence [38].

**Figure 4 ijms-25-08859-f004:**
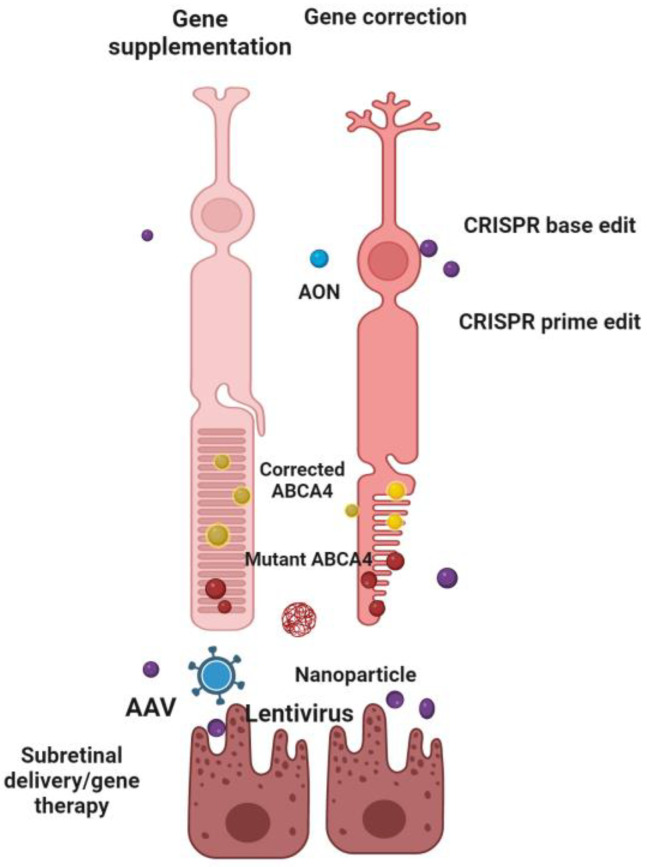
Gene therapy for *ABCA4* mutations: gene supplementation using viral vectors to deliver functional *ABCA4* genes, and gene correction through CRISPR and AONs to precisely edit or correct the existing mutations in retinal cells. Created with Biorender.

**Table 1 ijms-25-08859-t001:** Novel gene therapy approaches for Stargardt disease.

Strategy/ Intervention	Sponsor/Name of Study	Therapy	Clinical Trial/Website	Timeline	Phase
Gene therapy					
Gene supplementation	Ocugen/ GARDian West China Hospital	AAV (OCU410ST) AAV (JWK006)	NCT05956626 NCT06300476	2023/2025 2023/2026	Phase 1/2 Phase 1/2
Gene supplementation.	Sanofi	Lentivirus (SAR422459)	NCT01367444 NCT01736592	2011/2019 2012/2033	Phase 1/2 Phase 1/2
Gene supplementation	Splice Bio AAVantgarde Bio	Dual AAV	https://splice.bio/pipeline/; accessed on 26 June 2024 https://www.aavantgardebio.com/technology-pipeline/; accessed on 15 June 2024	expected 2024/2025	Phase 1/2
Gene modulation	-	AON	N/A	N/A	N/A
Gene editing	Ascidian Therapeutics/ STELLAR	ACDN-01	NCT06467344	2024/2030	Phase 1/2
Gene editing	-	CRISPR	-	N/A	N/A
Gene coding	Saliogen	SGT-1001	https://www.saliogen.com/pipeline/; accessed on 20 May 2024	expected 2024	N/A
Optogenetic therapy	Nanoscope Therapeutics/ STARLIGHT	vMCO-010	NCT05417126	2022/2023	Phase 2
	Raytherapeutics	RTx-021	https://raytherapeutics.com/pipeline/; accessed on 2 June 2024	N/A	N/A

### 4.2. Cell Replacement Therapy

With RPE and photoreceptor cells atrophy leading to decreased visual acuity for those in the final stages of disease, the therapeutic approaches discussed thus far would only be effective in avoiding additional vision impairment. However, one important problem with RPE transplantation is that it fails to directly address the etiology of the disease, and thus may not have beneficial effects over time. Providing additional induced pluripotent stem cells (iPSC)-derived RPE cells could only serve as a short-term remedy, especially since these cells, like the original cells, are prone to A2E and lipofuscin buildup [46]. 

Stem cell therapy represents one of the most promising methods for treating retinal dystrophies. Multiple investigations have been carried out to assess the safety as well as effectiveness of stem cell treatment for Stargardt disease (Table 2). In addition, different stem cell types and transportation strategies have been investigated. There are two main types of stem cells likely to be used for stem cell therapy: human embryonic stem cells (hESCs) and iPSCs, which are engineered in the lab. Lu et al. [47] demonstrated the long-lasting efficacy of hESC-derived RPE cells in immunodeficient rodents. Given the encouraging outcomes of stem cell treatment in animal models, human research followed and subsequent studies have investigated cell replacement therapy for retinal diseases [48,49]. These findings prompted Phase I/II clinical studies (NCT01625559 and NCT01345006), during which individuals with Stargardt’s disease underwent hESC-RPE transplants. Follow-up trials to these are underway (NCT02445612 and NCT02941991) and will be particularly important in confirming the safety of this treatment. A recent study published findings from patients with Stargardt disease who received autologous bone marrow-derived stem cells in each eye (NCT01920867) [50]. Although the study included STGD1, STGD3, and STGD4 patients, the genetic variation in each individual was not defined.

Ocata Therapeutics performed a research experiment using hESCs in patients with late-stage Stargardt disease. Initial findings have proven positive, with an effective safety trial held in the United States and the United Kingdom. Astellas Pharma has since acquired Ocata Therapeutics, which is developing a new trial in Stargardt’s illness using an enhanced RPE cell line. Their main efforts are towards geographic atrophy caused by age-related macular degeneration and Stargardt disease, using hESC-derived RPE (MA09-hRPE) cells and is now in Phase 1b clinical trials (NCT01345006). Opsis Therapeutics is planning a research study in which a patch made of retinal cells produced from stem cells will be used to restore vision. BlueRock’s ophthalmology cell therapy initiative, OpCT-001, attempts to restore vision lost due to this condition by replacing altered retinal tissue with functioning cells. OpCT-001 is an iPSC-derived cell therapy candidate being tested for the treatment of primary photoreceptor disorders.

### 4.3. Pharmacological Therapy 

Several clinical trials are now underway, targeting medications that target critical pathogenic processes in Stargardt disease (Table 3). The goals of these therapies are to reduce the production of harmful components of the retinoid cycle through lowering vitamin transportation or inhibiting various enzymes involved in the cycle or to specifically target harmful byproducts such as A2E or pathways activated by these metabolites (such as the complement cascade) (Figure 5).

As the precursor to 11-cis-retinol, vitamin A (all-trans-retinol) is frequently targeted during the visual cycle. It has been suggested that lowering the level of vitamin A may restrict the synthesis of all-trans-retinal/PE and, in turn, A2E. While regular vitamin A supplements may be harmful, as the *ABCA4* KO model shows [51], giving deuterated vitamin A was discovered as a viable alternative. A chemically-altered form of vitamin A that can be taken orally once a day has been shown to reduce retinal damage among individuals with Stargardt disease. Gildeuretinol (ALK-001) is intended to lower the propensity of vitamin A dimerization, the process by which molecules bond together to form a dimer, accumulating byproducts and lipofuscin in the retina. This procedure has the ability to reduce or stop the progression of retinal degenerative lesions. The TEASE trial (NCT02402660) is one of the first FDA-regulated Phase 2 multicenter clinical trials to evaluate a novel oral drug for retinal degeneration caused by vitamin A dimerization [52].

The primary retinol carrier, retinol-binding protein 4 (RBP4), the only protein that carries retinol from the liver to the eye, and has its plasma concentrations reduced by STG-001, an indirect visual cycle modulator that also slows down the visual cycle and the build-up of cytotoxic retinoids [53]. In addition to Stargardt disease, RBP4 is associated with obesity, insulin resistance, and cardiovascular disorders [52]. RBP4 is necessary for the delivery of retinol to the RPE, and the RPE expresses a particular RBP4 receptor (STRA6) that controls the absorption of vitamin A. Since they do not express the RBP4 receptor, other extrahepatic tissues do not require transport of retinol bound to RBP4 [54]. There are presently a few RBP4 antagonists being investigated. Currently, a Phase 2a clinical study examining two distinct dosages of STG-001 is being conducted on individuals with Stargardt disease (NCT04489511). Tinlarebant (LBS-008) is a novel oral medication that functions by lowering and preserving serum levels of RBP4. Tinlarebant inhibits the production of these toxins by adjusting the quantity of retinol that reaches the eye. Tinlarebant has shown target selectivity and potency in clinical trials, which may have therapeutic implications for treating geographic atrophy (GA) and STGD1 patients. In addition to undertaking a Phase 3 study (DRAGON), a Phase 2/3 study (DRAGON II), and a Phase 3 study (PHOENIX) (NCT05949593) in subjects with GA, the research team has finished a two-year Phase 2 investigation of Tinlarebant in teenage STGD1 participants [55,56].

The Stargardt Remofuscin Treatment Trial (STARTT) was a 2-year study comparing Remofuscin tablets to a placebo in individuals with Stargardt disease. The active component of Remofuscin, Soraprazan, underwent Phase I and Phase II clinical trials and was initially developed as an acid pump antagonist to treat gastro-esophageal reflux disease. Remarkably, lipofuscin of the RPE was also reported to be removed by Soraprazan in investigations conducted on non-human subjects. There is strong evidence to suggest Remofuscin as a viable medication candidate for the treatment of Stargardt disease [57,58,59].

Increased oxidative stress is thought to be an important variable in many ocular diseases that cause degeneration and visual loss [60]. Many of them develop with similar basic pathways to other central nervous system degenerative disorders, such as cell death, inflammation, and oxidative stress [61]. For the treatment of this group of diseases, a number of medications having anti-inflammatory and/or antioxidant qualities have demonstrated encouraging outcomes both in vivo and in vitro [62]. N-acetylcysteine amide (NACA) GMP-grade NPI-001 is an investigational antioxidant medication. NPI-001 retained photoreceptor cells and functionality in preclinical animal experiments. Nacuity’s GMP-grade NPI-001 solution underwent a successful Phase 1 clinical trial in a group of healthy participants, with no major adverse events reported. The SLO-RP study, a multicenter Phase 1/2 clinical trial of Nacuity’s patented NPI-001 tablets, has completed its target enrollment consisting of patients with Usher syndrome-associated RP (NCT04355689). Although this study focuses on Usher syndrome, the shared pathophysiology among all IRDs suggests that the findings could potentially be applied to Stargardt disease in the future.

The immunological response relies heavily on the complement cascade. Avacincaptad pegol (Zimura) is an aptamer that has the ability to block complement factor C5 function [63]. In conditions where the complement system is overactive or dysregulated, such as some IRDs, proteolytic C5 activation inhibition is a viable treatment strategy [64]. It is possible to prevent or slow down the advancement of autosomal recessive Stargardt disease by inhibiting complement C5. The possible safety benefit of reducing the upstream complement inhibition in the eye may also come from C5′s downstream placement within the complement cascade [65]. One study is currently active and aims to assess the safety and effectiveness of an intravitreal avacincaptad pegol injection versus a placebo in individuals with STGD1 (NCT03364153).

Emixustat is a very small molecule that pioneered a new family of chemicals known as visual cycle modulators. Emixustat hydrochloride is a non-retinoid molecule that inhibits retinoid isomerohydrolase (RPE65), a protein specific to the RPE. It is hypothesized to inhibit visual chromophore production and avoid the buildup of damaging retinal byproducts [66]. In animal models of Stargardt disease, emixustat has been shown to reduce A2E accumulation while protecting the retina from light-induced damage [67]. In 2018, Kubota Vision launched a Phase 3 clinical research for Stargardt disease. The primary purpose of this trial is to investigate if emixustat decreases the extent of macular atrophy development when compared to a placebo (NCT03772665).

Fenretinide, also known as N-(4-hydroxyphenyl) retinamide (4-HPR), is a synthetic derivative of retinoic acid. It competes with retinol to bind with RBP4 and reduces the levels of vitamin A derivatives, particularly retinol and retinyl esters, which are crucial for the visual cycle. Mata et al. investigated the comparison between fenretinide and a placebo (NCT00429936), but there was inconsistent reporting of results and adverse events [68].

Metformin, a drug primarily used to treat type 2 diabetes, has recently attracted interest for its potential role in treating Stargardt disease, a genetic retinal disorder that leads to progressive vision loss. The connection between metformin and Stargardt disease is based on the drug’s broader biological effects, which include anti-inflammatory properties, modulation of oxidative stress, and the potential to impact cellular metabolism. A group of researchers used mass spectrometry on treated animals’ eyes, which showed that metformin was transported to target tissues. FAF analysis revealed that the experimental group accumulated less lipofuscin. Their investigation demonstrates that medications targeting lipid processing abnormalities can be examined for their potential to decrease the progression of Stargardt disease [69]. While these mechanisms suggest potential benefits, clinical evidence supporting metformin’s effectiveness in Stargardt disease is still limited.

Other therapies have previously been studied, such as direct visual cycle inhibitors such as Accutane-Isotretinoin [70] and Antizol-Fomepizole (NCT00346853) [71], which prevent all-trans retinol from being oxidized to all-trans-retinal. Furthermore, VM200, a primary amine that interacts with all-trans-retinal to produce a non-toxic Schiff base, inhibits the development of A2E and cellular damage. VM200 was studied in preclinical trials and demonstrated positive outcomes in murine models [5,72].

### 4.4. Dietary Supplements

Dietary supplements, particularly those rich in antioxidants such as omega-3 fatty acids, lutein, and zeaxanthin, have been explored for their potential role in managing Stargardt disease (Table 4). These supplements are believed to help reduce oxidative stress in the retina, which may slow the progression of retinal degeneration in patients with this condition. However, while some studies suggest benefits, more research is needed to establish their effectiveness in Stargardt disease management. In the STARSAF02 experiment (NCT01278277), saffron intake has been investigated for Stargardt disease [73]. Although the drug was easily tolerated, there was no change in vision. The scientists discovered that long-term supplementation contributed to maintaining visual function after following some individuals for 36 months outside of the clinical trial scan period, but these findings need to be confirmed. Alongside its antioxidant activity, saffron has a complex mode of action that also promotes tissue resilience mechanisms, providing neuroprotection against oxidative damage [74].

Fatty acids play a significant role in the pathophysiology and potential treatment of Stargardt disease. Omega-3 and omega-6 fatty acids, in particular, are of interest due to their anti-inflammatory and neuroprotective properties. Omega-3 fatty acids, such as eicosapentaenoic acid (EPA) and docosahexaenoic acid (DHA), have been shown to support retinal health and may help mitigate the progression of retinal degenerative diseases by reducing oxidative stress and inflammation. Conversely, a high intake of omega-6 fatty acids, which are found in many processed foods, can promote inflammation and potentially exacerbate retinal damage. The balance between these fatty acids is crucial; a diet rich in omega-3s and low in omega-6s is believed to support better retinal health [75,76]. These fatty acids are used in therapeutic strategies for Stargardt disease to leverage their protective effects on retinal cells, potentially slowing disease progression and preserving vision. A recent study has shown that oral omega-3 fatty acid supplementation (3.7 g; EPA–DHA = 5:1) can enhance both objective and subjective vision in people with dry age-related macular degeneration and Stargardt disease [76]. Very-long-chain polyunsaturated fatty acids (VLC-PUFAs) are not present in normal animal diets, but they are found exclusively in the retina and among other tissues, implying the biological significance of their biosynthesis from shorter-chain dietary precursors through the activity of the ELOVL4 enzyme. While advancement has been achieved by analyzing human donors’ retinal samples and investigating animal models with ELOVL4 deficiency, the absence of pure VLC-PUFAs has impeded future physiological and therapeutic research. While a research group has made progress in the study of VLC-PUFAs by manufacturing one of them in large enough quantities for small-animal testing, much more work remains to be carried out [77]. Two months of omega-3 supplementation (blood arachidonic acid/eicosapentaenoic acid ~1.0–1.5) in rodents decreased lipofuscin granule formation in the retina and preserved the photoreceptor layer, indicating that omega-3 supplementation delays typical age-related degenerative changes in the retina [78].

### 4.5. Future Perspectives in Therapy 

Novel clustered regularly interspaced palindromic repeats(CRISPR)-based molecular techniques are also being investigated as a potential treatment for Stargardt disease [11]. Lately CRISPR/Cas9 gene editing was successfully used to fix pathogenic *ABCA4* mutations in human iPSCs [79]. The efficacy of CRISPR-Cas9 gene editing could differ according to the location and nature of the mutation, and off-target effects might result in unexpected changes and potentially damaging consequences. There are certain security issues with the gene editing approaches being researched for Stargardt, including the insertion of double-stranded breaks in the genome during editing. Gene editing techniques generate double-stranded breaks, which have the potential to activate error-prone endogenous DNA repair processes, resulting in unintended consequences [80]. The incorporation of hiPSCs with CRISPR-Cas9 technology allows for in vitro gene editing in patient-derived cells to fix particular mutations and allow maturation into retinal cells for autologous transplantation, constituting a potent therapy [79]. The results of one research study using the CRISPR technique are promising, as the scientists were successful in efficiently repairing the mutation in the *ABCA4* gene in patient-derived hiPSCs. The use of hiPSCs, which are derived from the patient’s cells, represents an intriguing possibility for individualized healthcare because it avoids the risk of immune rejection and enables the discovery of patient-specific medicines [81].

## 5. Conclusions

Stargardt disease is among the most frequent IRDs and has an extremely diverse clinical and genetic background. Throughout the last few years, extensive medical and genetic characterization has enhanced our understanding of the disease processes, biology, and outcomes, allowing for several treatment studies. Scientists and doctors have investigated a variety of therapy techniques, with encouraging pre-clinical results yet to be tested in humans. CRISPR-based techniques may be the most successful future treatments. The meticulous pathogenicity assessment of *ABCA4* variations, as well as a knowledge of their repercussions, are critical for creating associations between genotype and phenotype and mutation-dependent therapeutics. Combining existing therapy with modern ophthalmological procedures may improve eyesight preservation or restoration. With sustained work, effective therapies for Stargardt disease are expected in the near future.

## Figures and Tables

**Figure 1 ijms-25-08859-f001:**
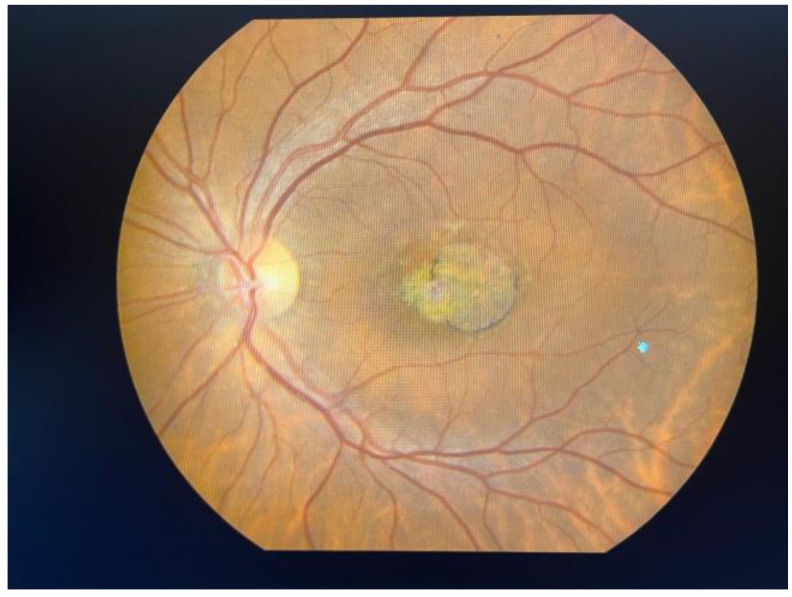
Left eye color fundus image of a 52-year-old patient with Stargardt disease. Personal casuistry.

**Figure 2 ijms-25-08859-f002:**
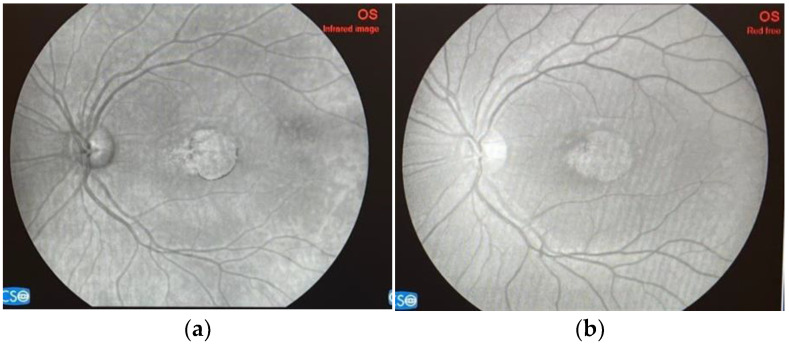
Left eye fundus image of the same patient. (**a**) Infrared image; (**b**) Red-free image.

**Figure 3 ijms-25-08859-f003:**
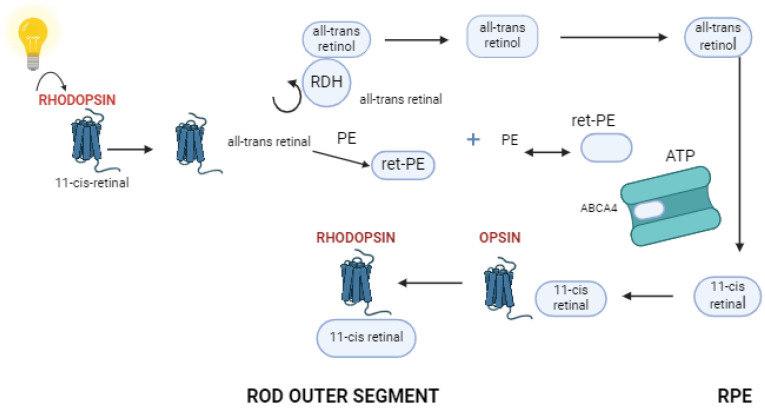
Presentation of the retinoid cycle. When rhodopsin (11-cis retinal) becomes stimulated by a photon, it undergoes a molecular conversion into all-trans-retinal, a process that occurs in the disk membrane of photoreceptor outer segments. The recently produced all-trans retinal must next be transferred over the disk membrane and into the cytoplasm. Some of the all-trans retinal diffuses straight to that subcellular region, but a large portion diffuses into the lumen of the disk membrane, where it interacts with phosphatidylethanolamine (PE) to generate N-retinylidene-PE (retPE), which remains trapped due to its protonated condition. *ABCA4* actively moves ret-PE across the disk membrane. On the cytoplasmic side, ret-PE separates into all-trans retinal and PE. All-trans retinal is then converted into all-trans retinol by all-trans retinol dehydrogenase (RDH). Afterward, all-trans retinol is delivered to the retinal pigmented epithelium (RPE), where the cycle continues, as shown. Created with Biorender.

**Figure 5 ijms-25-08859-f005:**
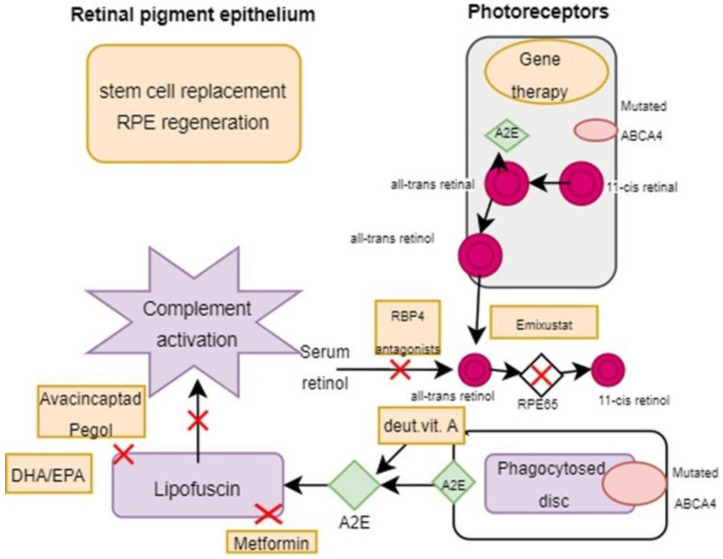
Therapeutic strategies targeting retinal pigment epithelium (RPE) and photoreceptors in Stargardt disease. This figure illustrates key therapies: gene and stem cell therapy for *ABCA4* mutation correction and RPE regeneration, deuterated vitamin A and RBP4 antagonists to reduce toxic retinoid byproducts, emixustat to inhibit harmful visual cycle enzymes, metformin and avacincaptad pegol to decrease lipofuscin and inhibit the complement cascade, and DHA/EPA to reduce oxidative stress and inflammation. Created with Draw.io (https://app.diagrams.net/ (accessed on 15 June 2024).

**Table 2 ijms-25-08859-t002:** Novel gene stem cell replacement approaches for Stargardt disease.

Strategy/ Intervention	Sponsor/Name of Study	Therapy	Clinical Trial/Website	Timeline	Phase
Stem cell therapy	Federal University of São Paulo Astellas	hESC-RPE	NCT02903576 NCT02941991	2015/2019 2013/2019	Phase I/II -
	MD Stem Cells/ SCOTS	BMSC	NCT01920867	2012/2020	-
	CHABiotech Astellas	MA09-hRPE	NCT01625559 NCT01345006 NCT01469832 NCT02445612	2012/2015 2011/2015 2011/2015 2012/2019	Phase I Phase I/II Phase I/II N/A
	BlueRock Therapeutics/FUJIFILM Cellular Dynamics/Opsis Therapeutics	iPSC (OpCT-001)	-	N/A	N/A

**Table 3 ijms-25-08859-t003:** Novel drug therapy approaches for Stargardt disease.

Strategy/ Intervention	Sponsor/Name of Study	Therapy	Clinical Trial/Website	Timeline	Phase
Drug therapy					
Deuterated vitamin A	Alkeus Pharmaceuticals TEASE	ALK-001	NCT02402660 NCT02230228 NCT04239625	2015/2025 2014/2015 2019/2026	Phase 2 Phase 1 Phase 2
RBP4 inhibition	Belitebio/ DRAGON DRAGON II PHOENIX	Tinlarebant	NCT05244304 NCT06388083 NCT05949593	2022/2025 2024/2027 2023/2027	Phase 3 Phase 2/3 Phase 3
	Stargazer Pharmaceuticals	STG-001	NCT04489511	2020/2021	Phase 2
	Sirion Therapeutics	Fenretinide	NCT00429936	2006/2010	Phase 2
	Stargazer Pharmaceuticals	A1120	-	N/A	N/A
Removal of lipofuscin	Katairo/ STARTT	Soraprazan (Remofuscin)	EudraCT No. 2018-001496-20	2019/2023	Phase 2
C5 inhibition	Astellas Pharma	Avacincaptad pegol (Zimura)	NCT03364153	2018/2025	Phase 2
Visual cycle modulator	Kubota Vision/seaSTAR	Emixustat	NCT03033108 NCT03772665	2017/2017 2019/2022	Phase 2 Phase 3
Increase macroautophagy	National Eye Institute	Metformin	NCT04545736	2020/2027	Phase 1/2

**Table 4 ijms-25-08859-t004:** Novel dietary supplements for Stargardt disease.

Strategy/ Intervention	Sponsor/Name of Study	Therapy	Clinical Trial/Website	Timeline	Phase
Diet					
	Ophthalmos Research and Education Institute/ MADEOS	Omega-3 fatty acids	NCT03297515	2019/2020	Not applicable
	Catholic University of the Sacred Heart/ STARSAF02	Saffron	NCT01278277	2011/2017	Phase 1/2
	National Eye Institute	DHA	NCT00060749	2003/2007	Phase I

## Data Availability

No new data were created.

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
