# Peer review of "Emerging Therapeutic Approaches and Genetic Insights in Stargardt Disease: A Comprehensive Review"

_ijms, 2024, doi:10.3390/ijms25168859_

Round 1

Reviewer 1 Report

Comments and Suggestions for Authors

This review focuses on new approaches to treat the Stargardt disease, a pathology due to the retinal pigment epithelium malfunction and subsequent photoreceptor degeneration. Overall, the manuscript is well structured and I only suggest some minor revisions:

Acronyms should be explained at their first appearance.

For example: Line 81 STGD; Lines 439 and 441 GA; Line 462 Is RP retinitis pigmentosa?

 The list of references is sufficiently wide, although some more citations might be included. For example (Line 151), it should be mentioned that RPE have also neuro-inductive ability, as reported not only on photoreceptors but also on adult stem cells (doi: 10.1002/jnr.21813 - doi: 10.4252/wjsc.v13.i11.1783)

The two figures should be better described in the legends. 

In the table, line subdivisions for each approach should be checked.

Comments on the Quality of English Language

Sufficiently good.

Author Response

Dear Reviewer

Thank you for your detailed feedback. We have made the following revisions in response to your suggestions:

1. We have ensured that all acronyms throughout the manuscript are explained when they first appear in it 

2. We have added citations to support the neuro-inductive abilities of the retinal pigment epithelium , referencing studies suggested that demonstrate its impact on both photoreceptors and adult stem cells.

3. The legends for the figures have been expanded to provide a clearer and more comprehensive description of the content and significance of each figure. This includes detailing the mechanisms and pathways illustrated, ensuring the figures are fully understandable on their own.

4. We have reviewed and corrected the table line subdivisions to ensure consistency and clarity. Each approach within the table is now clearly delineated, making the information easier to follow. We also subdivided the table into 4 different sections.

Once again, thank you for your valuable suggestions.

Best regards,

The authors.

Reviewer 2 Report

Comments and Suggestions for Authors

Ghenciu and coauthors mad a very comprehensive review about the current therapeutic approaches of Stargardt disease. There are several comments about this paper

1. Line 41, german should be corrected to German.

2.  Diagrams to demonstrated the fundus pictures of Stargardt disesse, including color photo and FAF will be  good for the readers.

3. A diagram to show the structure of ABCA4 gene and protein and the metabolic pathway of retinoic acid between photoreceptor and RPE cells will be needed.

4. Line 143-162, These 2 paragraphs are redundant and could be mote concise. 

4. Figure 2 could be modified , for example the function of metformin was  not mentioned in the article.

5. Table 1 looks rather complicated.   The table could be divided into several tables and put into the corresponding sections and the read will be more easy to read. 

Comments on the Quality of English Language

The English is fine.  Minor  revision will be needed. 

Author Response

Dear Reviewer,

First, we would like to thank you for your valuable feedback and suggestions, which have significantly enhanced the quality of our manuscript.

1. We have corrected "german" to "German"

2. As per your recommendation, we have added images that demonstrate the fundus pictures characteristic of Stargardt disease, including both color photos and infrared/red free images. Unfortunately, we do not have images with FAF from our own personal case.

3. We have included a new diagram illustrating the metabolic pathway of retinoic acid between photoreceptor cells and the retinal pigment epithelium. 

4. The paragraphs in Lines 143-162 have been revised for conciseness to eliminate redundancy. 

5. Metformin was already mentioned in Table 1. To further address your comment, we have added a paragraph detailing metformin’s role in Stargardt disease. Figure 2 has also been updated to include this information and also better explain all the mechanisms

6. We have restructured Table 1 by dividing it into smaller tables, placing them in the corresponding sections of the manuscript. 

Thank you once again for your insightful comments and suggestions, which have greatly contributed to improving the manuscript. We believe these revisions have strengthened the overall quality and clarity of our work.

Best regards,

The authors